# *Ag4CL3* Related to Lignin Synthesis in *Apium graveolens* L.

**Xiu-Lai Zhong** [1,†], **Shun-Hua Zhu** [1,†], **Qian Zhao** [1,2], **Qing Luo** [1], **Kun Wang** [1], **Zhi-Feng Chen** [3,*]
**and Guo-Fei Tan** [1,4,*]

1    Institute of Horticulture, Guizhou Academy of Agricultural Sciences/Horticultural Engineering Technology
     Research Center of Guizhou, Guiyang 550006, China; gzzyzxl@foxmail.com (X.-L.Z.);
     zsh2801@163.com (S.-H.Z.); lqing985@foxmail.com (Q.Z.); wangkunfun@hotmail.com (Q.L.);
     zhaoq217@163.com (K.W.)
2    Key Laboratory of Plant Resource Conservation and Germplasm Innovation in Mountainous Region (Ministry
     of Education), College of Life Sciences/Institute of Agrobioengineering, Guizhou University,
     Guiyang 550025, China
3    College of Biology and Agricultural Technology, Zunyi Normal College, Zunyi 563006, China
4    Faculty of Agronomy, Jilin Agricultural University, Changchun 130118, China
*    Correspondence: cxf810@163.com (Z.-F.C.); tagfei@foxmail.com (G.-F.T.)
†    These authors contributed equally to this work.

**Abstract:** 4-Coumarate: coenzyme A ligase (4CL; EC 6.2.1.12) is an important enzyme in the phenyl-propanoid metabolic pathway that controls the biosynthesis of lignin and flavonoids. In this study, to identify the function of the *Ag4CL3* gene of celery, the *Ag4CL3* gene was cloned from celery cv. "Nanxuan Liuhe Ziqin". Sequence analysis results showed that the *Ag4CL3* gene contained an open reading frame (ORF) with a length of 1688 bp, and 555 amino acids were encoded. The Ag4CL3 protein was highly conserved among different plant species. Phylogenetic analysis demonstrated that the 4CL proteins from celery and carrot belonged to the same clade. The Ag4CL3 protein was mainly composed of 31.89% α-helixes, 18.02% extended strands, 6.67% β-turns, and 43.42% random coils, and the signal peptide was unfound. A total of 62 phosphorylation sites and a class-I superfamily of adenylate-forming domains were found. As the growth time increased, the plant height and stem thickness also increased, and the petiole lignin content increased and became lignified gradually. The relative expression levels of the *Ag4CL3* gene in "Nanxuan Liuhe Ziqin" petioles were higher than those in other tissues, with the highest level occurring 70 d after sowing. The lignin contents in the transgenic *Arabidopsis thaliana* lines hosting the *Ag4CL3* gene were higher than those in the WT. In this study, the overexpression of *Ag4CL3* led to the significant upregulation of lignin biosynthesis gene expression in transgenic *A. thaliana* plants, except for *AtPAL, AtCCR,* and *AtLAC.* This study speculates that *Ag4CL3* genes are related to lignin synthesis in *A. graveolens*.

**Keywords:** *Ag4CL3*; *A. graveolens*; lignin; gene clone; expression

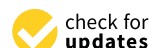



## 1. Introduction

Celery (*Apium graveolens* L.) is a widely cultivated Apiaceae vegetable worldwide [1], and its leaves and petioles are the main edible tissues. The medicinal value and rich nutrients of celery [2,3], such as dietary fiber [4], anthocyanin [5,6], and apigenin [7–9], have been reported. Dietary fiber is a polysaccharide that is not easily digested and includes cellulose, hemicellulose, and lignin [10].

Lignin is an important secondary metabolite and is the second largest biopolymer in the world. In particular, lignin accounts for 30% of the organic carbon content in the biosphere [11]. Moreover, lignin is a complex phenolic polymer that is composed of p-coumaryl alcohol, coniferyl alcohol, and sinapyl alcohol, which are three alcohol monomers [12]. Its structure is relatively stable and it is one of the main components that make up the skeleton of woody and herbaceous plants. Monomers in the chemical composition of lignin have three major types: p-hydroxyphenyl (H) lignin, formed by

the polymerization of p-hydroxyphenylpropane structural monomers; guaiacyl (G) lignin, formed by the polymerization of guaiacyl propane structural monomers; and syringal (S) lignin, formed by the polymerization of syringal propane structural monomers [13].

Lignin is an insoluble dietary fiber that naturally exists in vegetables and fills the cellulose frame. It is an important physiological function that enhances the mechanical strength of plants, assists in the transport of water in tissues, and improves the resistance of plants to stress [12]. The lignin content dynamically changes during plant growth and development, the vegetable tissue gradually becomes solid and rough, and the tissue becomes lignified and not easy to chew, which seriously reduces vegetable quality and affects taste [14].

The 4-Coumarate acid: coenzyme A ligase (4CL; EC 6.2.1.12) is related to lignin biosynthesis [15], which can catalyze six phenolic acids, namely, coumaric acid, cinnamic acid, caffeic acid, ferulic acid, 5-hydroxy ferulic acid, and sinapic acid, to generate corresponding CoA lipids: cinnamoyl CoA, coumaryl CoA, caffeyl CoA, feruloyl CoA, 5-hydroxyl feruloyl CoA, and sinapol CoA. These CoA lipids are precursors of lignin synthesis, and coumarin coenzyme A is a common precursor of lignin and flavonoid synthesis [16]. Therefore, 4CL is at the turning point of the phenylpropanoid metabolic pathway to form different types of products, which is related to the trend of the lignin-specific pathway [17].

In previous research, *4CL* genes have been cloned and studied in a large variety of plants, where they comprise small gene families in most cases [18]. For example, parsley (*Petroselinum crispum*) has two *4CL* genes [19], tobacco (*Nicotiana tabacum*) and mulberry (*Morus alba*) have three *4CL* genes [20,21], the *Arabidopsis* genome [22] and soybean *(Glycine max)* [23] have four *4CL* genes, and the rice (*Oryza sativa*) genome has five *4CL* genes [24]. Phylogenetic analysis shows that *4CLs* in dicotyledonous plants can be divided into two categories: class I and class II. Class-I *4CLs* are mainly involved in the biosynthesis of lignin, whereas class-II *4CLs* are often involved in pathways other than lignin in the phenylpropanoid pathway. For example, in *Arabidopsis*, *At4CL1* and *At4CL2* mainly participate in the synthesis of lignin, whereas *At4CL3* mainly affects the synthesis of flavonoids [25]. Similar to the situation in *Arabidopsis*, *Pt4CL1* is highly expressed in the xylem of *Populus euphratica*, which is mainly involved in lignin biosynthesis; *Pt4CL2* is expressed in the epidermis of leaves and stems, and participates in the biosynthesis of phenolic substances, such as flavonoids [26].

In celery, the function of the *Ag4CL* gene in lignin biosynthesis is unclear. In this study, we cloned the *Ag4CL3* gene from celery and detected its expression patterns in different tissues and growth stages of celery. Transgenic *A. thaliana* plants overexpressing *Ag4CL3* were obtained to examine the lignin level, and then comparison of lignin content between transgenic and wild type *Arabidopsis*. This study further identified the roles of *Ag4CL3* in lignin biosynthesis, laying a foundation for future studies on the regulation of lignin synthesis and breeding new cultivars.

## 2. Materials and Methods

### 2.1. Plant Material and Experimental Design

Celery cv. "Nanxuan Liuhe Ziqin", Columbia wild type *A thaliana* ecotype (WT), and transgenic *A thaliana* were grown in pots that contained a mixture of soil, vermiculite, and perlite at the Institute of Horticulture, Guizhou Academy of Agricultural Sciences (106.67° E, 26.51° N). The celery and *A thaliana* plants were grown at 25/18 °C (day/night) for 16/8 h in a phytotron with a relative humidity of 70~75%, and the light intensity was 300 μmol $m^{-2} \cdot s^{-1}$ during daytime. The petioles of celery were collected 30 d (seedling stage), 50 d (growth period), and 70 d (commercial stage) after sowing. The roots, leaves, and petioles of celery were collected 70 d (commercial stage) after sowing. All the samples were immediately frozen in liquid nitrogen and then stored at −80 °C for RNA extraction. WT (wild type) and *Ag4CL3*-OE lines of *A. thaliana* were grown on a chat/vermiculite/perlite mixture. Each experiment was performed in three biological replicates.

## 2.2. Determination of Lignin Content

The lignin contents of the purple celery petioles and *A. thaliana* rosette leaves were extracted and measured in accordance with previous research methods [27,28]. The sample was ground in liquid nitrogen with a mortar and pestle, approximately 1 g of the sample was ground in 99.7% ethanol, and the mixture was centrifuged at 14,000 rpm·min$^{-1}$ for 20 min. The sediment was collected and air-dried at room temperature overnight. Approximately 10 mg of the dried sediment was weighed and transferred to a 2 mL clean centrifuge tube. Next, 1 mL of 2 M HCl and 0.1 mL of thioglycolic acid were added. The tube with the mixture was incubated at 100 °C for 8 h, cooled on ice, and then centrifuged at 14,000 rpm/min for 20 min at 4 °C. The sediment was washed with 1 mL of deionized water and dissolved in 1 mL of 1 M NaOH. It was incubated at 25 °C for 18 h, and centrifuged at 14,000 rpm/min for 20 min, and the supernatant was transferred to a new 2 mL centrifuge tube. About 1 mL of concentrated HCl was added, and the mixture was kept at 4 °C for 6 h to precipitate the lignin thioglycolic acid. After centrifugation at 14,000 rpm/min for another 20 min, the sediment was dissolved in 1 mL of 1 M NaOH. Finally, the 1 M NaOH solution was used as a blank control, and the absorbance value was measured at 280 nm.

## 2.3. Total RNA Extraction and cDNA Synthesis

The total RNA of celery and *A. thaliana* was extracted using a total RNA extraction kit (Huayueyang, Beijing, China), according to the instructions. Then, the RNA was converted into cDNA using a HiScript III 1st Strand cDNA Synthesis Kit (+gDNA wiper) (Vazyme, Nanjing, China), based on the manufacturer's instruction.

## 2.4. Bioinformatics Analysis

The 4CL3 protein sequences of other species were downloaded from the National Center for Biotechnology Information (NCBI) database (https://www.ncbi.nlm.nih.gov/) (accessed on 24 November 2022). The primers were designed and obtained by Primer Premier 6.0. BioXM software, which was used to analyze the nucleotide and encoded amino acid sequences of the *Ag4CL3* gene. The secondary and tertiary structures of the Ag4CL3 protein were predicted and established using SOPMA software (http://pbil.ibcp.fr/) (accessed on 24 November 2022) and a SWISS models server (https://swissmodel.expasy.org/) (accessed on 24 November 2022), respectively. The conserved functional domain of the Ag4CL3 protein was predicted by the CDD of NCBI. To investigate the phylogenetic relationships of the 4CL proteins, MEGA7.0 software was used to construct a phylogenetic tree using the maximum likelihood method, Jones-Taylor-Thornton (JTT) model, Gamma Distributed (G). According to the 4CL classification of *Arabidopsis thaliana*, *Glycine max*, and *Populus tremuloides,* Ag4CL3 was classified and predicted.

## 2.5. Overexpression Vector Construction and A. thaliana Transformation

The putative *Ag4CL3* gene sequence was retrieved from the celery genome and transcriptome database [29,30]. The full length ORF (open reading frame) of *Ag4CL3* was amplified with specific primers (*Ag4CL3*-KpnI-F: 5′-GCGGGTCGACGGTACCATGCCAAGTC-TCAGCCAATC-3′; *Ag4CL3*-KpnI-R: 5′-TAGACATATGGGTACCTTATAATCTTGCTG-AGG CTCC-3′). The PCR product was cloned into pCAMBIA2301 and sequenced. The recombinant plasmid (35S: *Ag4CL3*) was introduced into *Agrobacterium tumefaciens* strain GV3101 via the electroporation method. The floral dip method was used for the *Agrobacterium-mediated* transformation of *A. thaliana* [31]. Transgenic *A. thaliana* was initially screened on MS medium containing kanamycin (50 mg·L$^{-1}$), and then PCR amplification and sequencing were conducted using *Ag4CL3* recombinant primers.

## 2.6. Real-Time Quantitative PCR Analysis

Real-time quantitative PCR (RT-qPCR) was conducted to detect the expression level of *Ag4CL3* in the seedling stage and different tissues of "Nanxuan Liuhe Ziqin". Premier 6.0 software was used to design primers (*Ag4CL3*-qF: 5′-ACTCTTCAGGCACTACTGGACGA-

3′, *Ag4CL3*-qR: 5′-CAGCATAAAAAAACC-AAACACAT-3′). The *AgActin* gene was used as an internal standard [32]. The specific primers of the samples of WT and transgenic *A. thaliana* plants for the lignin biosynthesis gene were also designed by Premier 6.0 software (Table 1). *AtActin* was used as an internal reference gene [33].

**Table 1.** The specific primers of transgenic *A. thaliana* plants about lignin-biosynthesis-gene.

| Gene | GenBank No. | Forward Primer (5′→3′) | Reverse Primer (5′→3′) |
|---|---|---|---|
| *AtActin* | NM_112764.4 | TCGCTGACCGTATGAGCAAAG | TGTGAACGATTCCTGGACCTG |
| *AtPAL* | NM_120505.4 | CGAGTAGTGACTGGGTGATGGA | AGGAGGGTGTTTACACGGATG |
| *AtC4H* | AM887619.1 | ACGGCGAGCATTGGAGGAAGA | TCTCCATAGTTATACTCAAAGC |
| *At4CL* | NM_104046.3 | CCGAATCTTTATTTCCACAG | CACCGTCACTTTACACCTCT |
| *AtCCR* | NM_001332191.1 | GTGCAAAGCAGATCTTCAGG | GCCGCAGCATTAATTACAAA |
| *AtCAD* | NM_119587.4 | TTGGCTGATTCGTTGGATTA | ATCACTTTCCTCCCAAGCAT |
| *AtHCT* | NM_124270.4 | GCCTGCACCAAGTATGAAGA | GACAGTGTTCCCATCCTCCT |
| *AtC3′H* | NM_128601.3 | GTTGGACTTGACCGGATCTT | ATTAGAGGCGTTGGAGGAT |
| *AtF5H* | NM_119790.3 | CTTCAACGTAGCGGATTTCA | AGATCATTACGGGCCTTCAC |
| *AtCCOAOMT* | NM_001342249.1 | CTCAGGGAAGTGACAGCAAA | GTGGCGAGAAGAGAGTAGCC |
| *AtCOMT* | NM_124796.4 | TTCCATTGCTGCTCTTTGTC | CATGGTGATTGTGGAATGGT |
| *AtPER* | NM_001344315.1 | CGGCGGTGTTGAAAGCGGT | GACATTATCCGCGAAACCATC |
| *AtLAC* | NM_125395.3 | GACCATACGTCACAGGTCAA | TGGTGTTATAAGGTAGAACC |

The ChamQ SYBR ColorMaster Mix (Vazyme, Nanjing, China) and Bio-Rad IQ5 real-time PCR system (Bio-Rad, Hercules, CA, USA) were used for qPCR. The qPCR used a 20 μL system, including 2 μL of cDNA, 0.5 μL of forward and reverse primers, 10 μL of SYBR Premix Ex Taq, and 7 μL of ddH$_2$O. Three biological replicates were performed for each sample. The reaction condition for qPCR is 95 °C for 5 min, 40 cycles at 95 °C for 10 s, 60 °C for 30 s, and melting curve analysis (65 to 95 °C, increasing by 0.5 °C every 5 s). The gene expression level was calculated by $2^{-\Delta\Delta CT}$ [34].

### 2.7. Statistical Analysis

All data in the text were obtained from the average of three biological repeats. Data significant differences were analyzed using SPSS 25 software by one-way ANOVA at 0.05 levels.

## 3. Results

### 3.1. Changes of Lignin Contents

As the growth time increased, the plant height and stem thickness also increased (Figure 1A–D and Table 2). Lignin content is an important index that influences the aging of celery. During the growth stage, the lignin content of the petioles gradually increased, and the lignin content of petioles at 70 d of growth increased to 222.25 mg·g$^{-1}$ (Figure 1D).

**Table 2.** Change of morphology of celery growth stage.

| Day after Sowing (d) | Plant Height (cm) | Stem Thick (cm) |
|---|---|---|
| 30 (seedling stage) | 13.45 ± 0.24 c | 1.64 ± 0.05 c |
| 50 (vigorous growth period) | 18.64 ± 0.27 b | 2.37 ± 0.10 b |
| 70 (commercial stage) | 26.33 ± 0.99 a | 3.06 ± 0.09 a |

Data are expressed as the means ± standard deviation (SD) of three replicates. Different lowercase letters indicate significant differences at 0.05 levels.

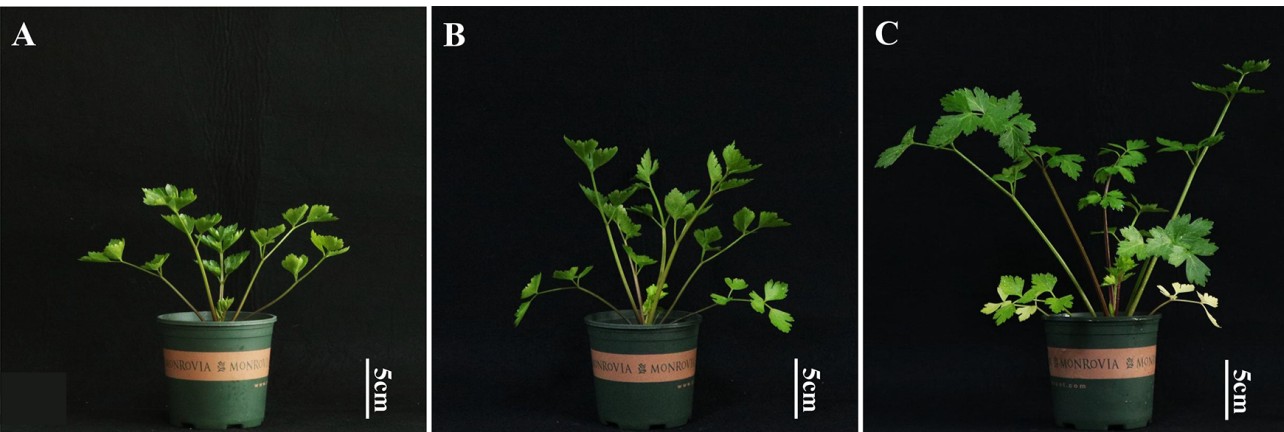

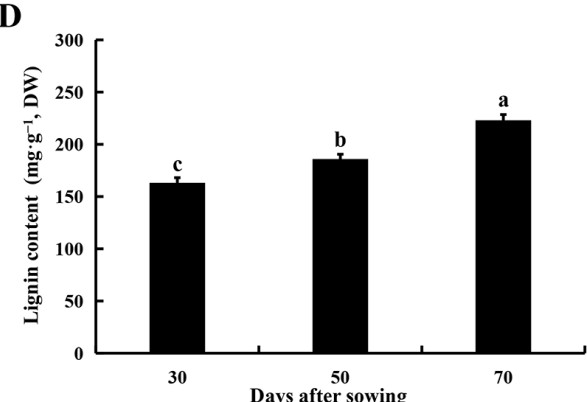

**Figure 1.** Morphological changes and lignin content of "Nanxuan Liuhe Ziqin" petioles during growth. (**A**) 30 d (seedling stage) after sowing; (**B**) 50 d (vigorous growth period) after sowing; (**C**) 70 d (commercial stage) after sowing; (**D**) Lignin content of celery during growth. Data are expressed as the means ± standard deviation (SD) of three replicates. Different lowercase letters indicate significant differences at 0.05 levels.

### 3.2. Analysis of the Ag4CL3 Sequence

The *Ag4CL3* gene cDNA sequence was successfully cloned from "Nanxuan Liuhe Ziqin" (GenBank OP156995) and contained an ORF length of 1668 bp, encoding 555 amino acids (Figure 2A). The theoretical isoelectric point was 6.89, and the protein molecular weight was 59.85 kDa. The secondary structure mainly consisted of 31.89% α-helixes, 18.02% extended strands, 6.67% β-turns, and 43.42% random coils (Figure 2B). The tertiary structure composition is consistent with its secondary structure composition (Figure 2C). The Ag4CL3 protein has a class-I superfamily of adenylate-forming domains. It has an active site, an AMP binding site, putative CoA binding site, and acyl-activating enzyme (AAE) conserved motif (Figure 2D). In order to investigate the phylogenetic relationships of 4CL proteins, a phylogenetic tree was constructed using the maximum likelihood method for a total of 10 amino acid sequences of 4CLs from *A. thaliana*, *G. max*, and *P. tremuloides* (Figure 3). The results showed that Ag4CL3 successfully clustered with At4CL1, At4CL2, Gm4CL1, Gm4CL2, and Pt4CL1, indicating that Ag4CL3 belonged to class I. Moreover, the functional studies of class I and class II genes in *A. thaliana*, *G. max*, and *P. tremuloides* showed that they were involved in lignin synthesis and flavonoid synthesis. Therefore, it is speculated that Ag4CL3 plays a pivotal role in lignin biosynthesis in the celery.

**A**

```
1      atgccaagtctcagccaatctccggggactattgatcccaacagcggcttcaactccgccacgagaacattccacagcctc
       M  P  S  L  S  Q  S  P  G  T  I  D  P  N  S  G  F  N  S  A  T  R  T  F  H  S  L
82     cgtccacctgttccactccctccaccatcacaaccattctccattactcaatatgcctcctctcttctcaactcctccacc
       R  P  P  V  P  L  P  P  P  S  Q  P  F  S  I  T  Q  Y  A  S  S  L  L  N  S  S  T
163    tctttctcccctgaaaccacttctttcctcatcgacgcagcctctgccactcgtgtttcttactctcagtttctcgaaaaa
       S  F  S  P  E  T  T  S  F  L  I  D  A  A  S  A  T  R  V  S  Y  S  Q  F  L  E  K
244    tctcagtctctctcggcctctctcctgtctaaattccccgagctttcgggccagaaagtagctcttattatttctccacct
       S  Q  S  L  S  A  S  L  L  S  K  F  P  E  L  S  G  Q  K  V  A  L  I  I  S  P  P
325    tcactggacctcccagtgctgtacttttctctcatttcgttaaacgtcattgtttctccagctaatccgctatctactccg
       S  L  D  L  P  V  L  Y  F  S  L  I  S  L  N  V  I  V  S  P  A  N  P  L  S  T  P
406    tctgaattggcacatcttgttgagttatgtaagcctgtgatcgcgtttgcaacgagttcagttgcgaaaaatcttccagct
       S  E  L  A  H  L  V  E  L  C  K  P  V  I  A  F  A  T  S  S  V  A  K  N  L  P  A
487    ctgccactcggaacaatcctgatggactcacctgagtttctgtcgatgtttaattcatcgaaaactaacgatgtacctaaa
       L  P  L  G  T  I  L  M  D  S  P  E  F  L  S  M  F  N  S  S  K  T  N  D  V  P  K
568    caaatttatatatctcagtccgatacagctgcaattctgtactcttcaggcactactggacgagtgaaaggagtggagtta
       Q  I  Y  I  S  Q  S  D  T  A  A  I  L  Y  S  S  G  T  T  G  R  V  K  G  V  E  L
649    actcaccggaacctgatcactgttattgctgatttgtattataacaagcgaacatcctcgggtgaggaagaatcaggtgtg
       T  H  R  N  L  I  T  V  I  A  D  L  Y  Y  N  K  R  T  S  S  G  E  E  E  S  G  V
730    gcgttgttcacgctgcctttgtttcatgtgtttggttttttttatgctgatcagaggatttgcattgggtgagaccttggtc
       A  L  F  T  L  P  L  F  H  V  F  G  F  F  M  L  I  R  G  F  A  L  G  E  T  L  V
811    ctgatggagagatttgatttgttgttaaaatgttagaggctgttgagaagtatagagttaattatatgcctgtttctcccccg
       L  M  E  R  F  D  F  V  K  M  L  E  A  V  E  K  Y  R  V  N  Y  M  P  V  S  P  P
892    atcgtggtggcgttggctaagtctgatctggtggccaagtatgatcttagctcactgaaattacttggttcaggtggagct
       I  V  V  A  L  A  K  S  D  L  V  A  K  Y  D  L  S  S  L  K  L  L  G  S  G  A
973    gcgcttggtagggagacttcggagaggttcactgctaggttcccaaatgtggaggtagctcagggctatggcatgactgag
       A  L  G  R  E  T  S  E  R  F  T  A  R  F  P  N  V  E  V  A  Q  G  Y  G  M  T  E
1054   actggggagggggcaactggaatgaacaaccaagaggagagtacgcggtatggatctggcggccgcttatctgccagcatc
       T  G  G  A  T  G  M  N  N  Q  E  E  S  T  R  Y  G  S  G  G  R  L  S  A  S  I
1135   gaaggtaaaatagttgatcctggaactggtaaggcccctaggacctggacaacaaggagagctatggttgcgaggacctaat
       E  G  K  I  V  D  P  G  T  G  K  A  L  G  P  G  Q  Q  G  E  L  W  L  R  G  P  N
1216   atcatgaaaggttatgtagcagataatgctgcaactgctgaaactttgacttctgatggctggttaaagactggtgacctc
       I  M  K  G  Y  V  A  D  N  A  A  T  A  E  T  L  T  S  D  G  W  L  K  T  G  D  L
1297   tgttactttgactcagatggcttcctgtatattgttgataggttgaaggagttgattaaatataaggcttaccaggttcct
       C  Y  F  D  S  D  G  F  L  Y  I  V  D  R  L  K  E  L  I  K  Y  K  A  Y  Q  V  P
1378   cctgccgagctagaacatataattcactcgattcctggagttgctgatgtagcagtaatcccatatcctgatgaagatgcg
       P  A  E  L  E  H  I  I  H  S  I  P  G  V  A  D  V  A  V  I  P  Y  P  D  E  D  A
1459   gggcagatacccatggcatatgtagtgaagtcctggaagcaatatatccgagagagagatccaagatttcgttgctaaa
       G  Q  I  P  M  A  Y  V  V  R  S  P  G  S  N  I  S  E  R  E  I  Q  D  F  V  A  K
1540   caggtatcaccatacaagaaggttcgacgtgttgcatttatcaacgctattccaaaatcacctgctggaaagatattaaga
       Q  V  S  P  Y  K  K  V  R  R  V  A  F  I  N  A  I  P  K  S  P  A  G  K  I  L  R
1621   agggaactggtgaatcatgctctttctggagcctcagcaagattataa
       R  E  L  V  N  H  A  L  S  G  A  S  A  R  L  *
```

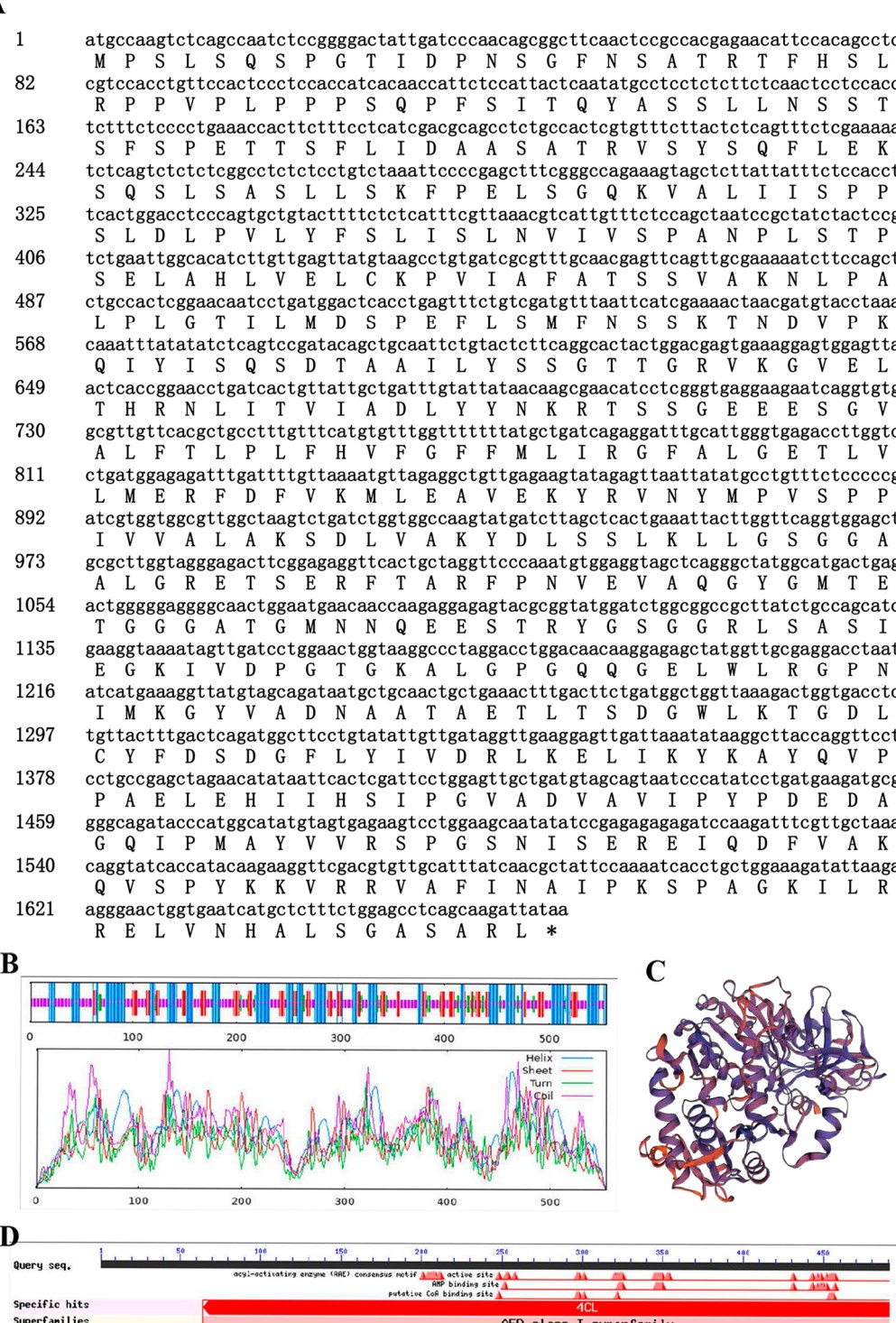

**Figure 2.** Bioinformatics analysis of Ag4CL3. (**A**) Nucleotide and encoded amino acid sequence of *Ag4CL3* gene, *: Stop codon; (**B**) The secondary structure model of the Ag4CL3 protein; (**C**) The tertiary structure model of the Ag4CL3 protein; (**D**) Prediction of conserved functional domain of theAg4CL3 protein.

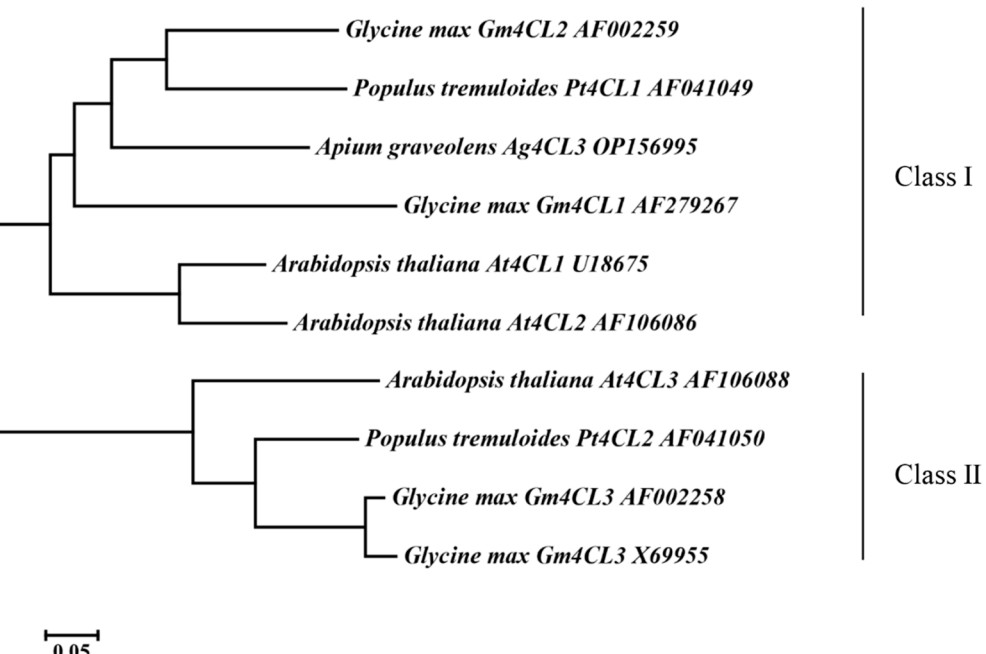

**Figure 3.** Phylogenetic tree of 4CL3 proteins from celery and other plant species.

*3.3. Ag4CL3 Gene Expression Analysis*

RT-qPCR was used to analyze the relative expression levels of the *Ag4CL3* gene in the different tissues of "Nanxuan Liuhe Ziqin". At 70 d after sowing, the *Ag4CL3* gene was expressed in the roots, leaves, and petioles of "Nanxuan Liuhe Ziqin", but the relative expression was significantly different. The relative expression of the *Ag4CL3* gene was highest in petioles, second in roots, and lowest in leaves (Figure 4A). In the seedling stage (30 d after sowing), the growth period (50 d after sowing), and the commercial stage (70 d after sowing), the relative expression of the *Ag4CL3* gene in the petioles showed remarkable differences. The highest relative expression of the *Ag4CL3* gene occurred at the commercial stage (70 d after sowing) (Figure 4B).

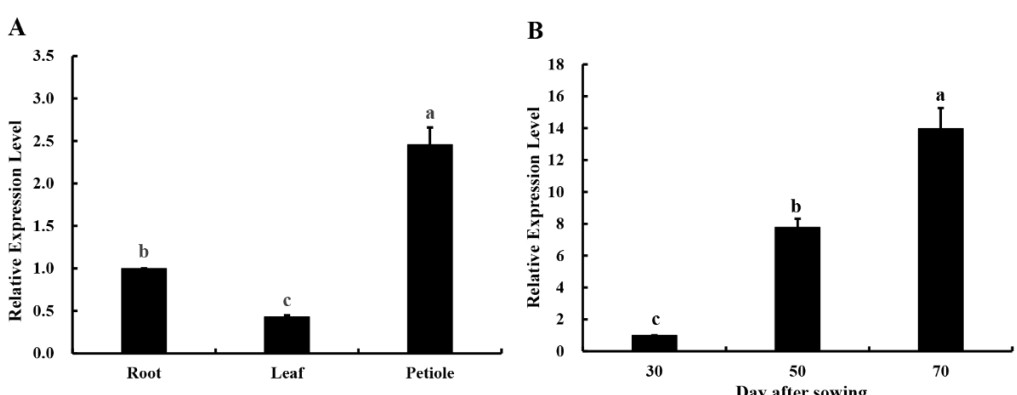

**Figure 4.** Comparison of the relative expression of the *Ag4CL3* gene in petioles at different growth stages (**A**) and in different tissues (**B**) of *A. graveolens* "Nanxuan Liuhe Ziqin". Data are expressed as the means ± standard deviation (SD) of three replicates. Different lowercase letters indicate significant differences at 0.05 levels.

*3.4. Identification of Transgenic A. thaliana and Overexpression of the Ag4CL3 Upregulated the Lignin Content in Arabidopsis*

To research the function of the *Ag4CL3* gene, transgenic *A. thaliana* lines were created via *Agrobacterium*-mediated transformation. The transgenic *A. thaliana* lines (L1, L2, and L3) were screened on MS medium containing kanamycin. Moreover, approximately 1600 bp

PCR products were observed in the transgenic lines L1, L2, and L3, based on PCR amplification, but they were not detected in the WT plants (Figure 5A). The results indicated that *Ag4CL3* was successfully transferred into *A. thaliana*, and three OE lines harboring the *Ag4CL3* gene were obtained. According to the lignin content determination, the transgenic *A. thaliana* lines exhibited higher lignin accumulation than the WT (Figure 5B). The lignin contents in the WT and transgenic *A. thaliana* lines were 106.57, 237.22, 250.10, and 316.55 mg·g$^{-1}$ DW (dry weight), respectively. The *Ag4CL3* gene was expressed significantly higher in transgenic *A. thaliana* than in WT plants (Figure 5C). These results indicate that the *Ag4CL3* gene was introduced, and that the successful overexpression of *Ag4CL3* upregulated the lignin level in transgenic *A. thaliana*.

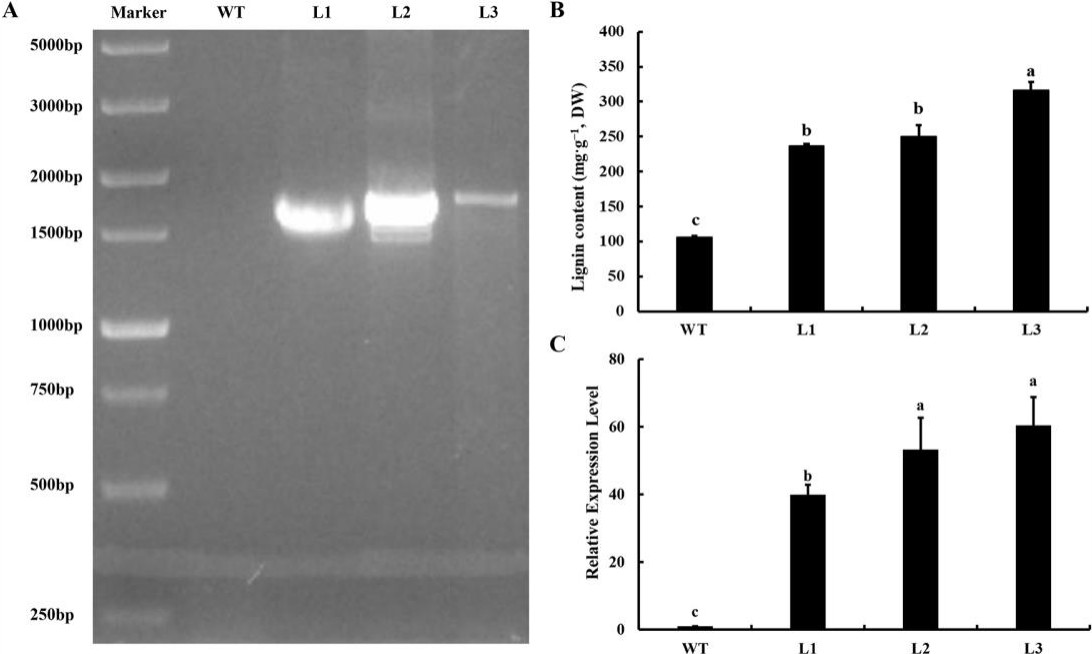

**Figure 5.** Identification of transgenic *A. thaliana* using (**A**) PCR amplification; (**B**) Comparison of lignin content between WT and transgenic *A. thaliana* lines (L1, L2, L3); (**C**) *Ag4CL3* gene expressed in WT and transgenic *A. thaliana* lines (L1, L2, L3). Data are expressed as the means ± standard deviation (SD) of three replicates. Different lowercase letters indicate significant differences at 0.05 levels.

### 3.5. Expression of Lignin Biosynthesis Related Genes in A. thaliana Plants

To investigate further the effect of *Ag4CL3* on lignin biosynthesis, the expression of other lignin biosynthesis genes was investigated in transgenic *A. thaliana* (Figure 6). The relative expression levels of *AtPAL* and *AtCCR* of the WT plants were higher than those of the transgenic *A. thaliana* plants, but the expression levels of the remaining genes in WT plants were lower than those in transgenic *A. thaliana* plants (L1, L2, L3). Furthermore, the expression levels of *AtHCT in* transgenic line L2 were higher than those in the WT and other transgenic lines. This result showed that gene expression in the transgenic *A. thaliana* plants was significantly upregulated, except for *AtPAL, AtCCR,* and *AtLAC.* Therefore, the overexpression of *Ag4CL3* can affect the biosynthesis of lignin in *A. thaliana*.

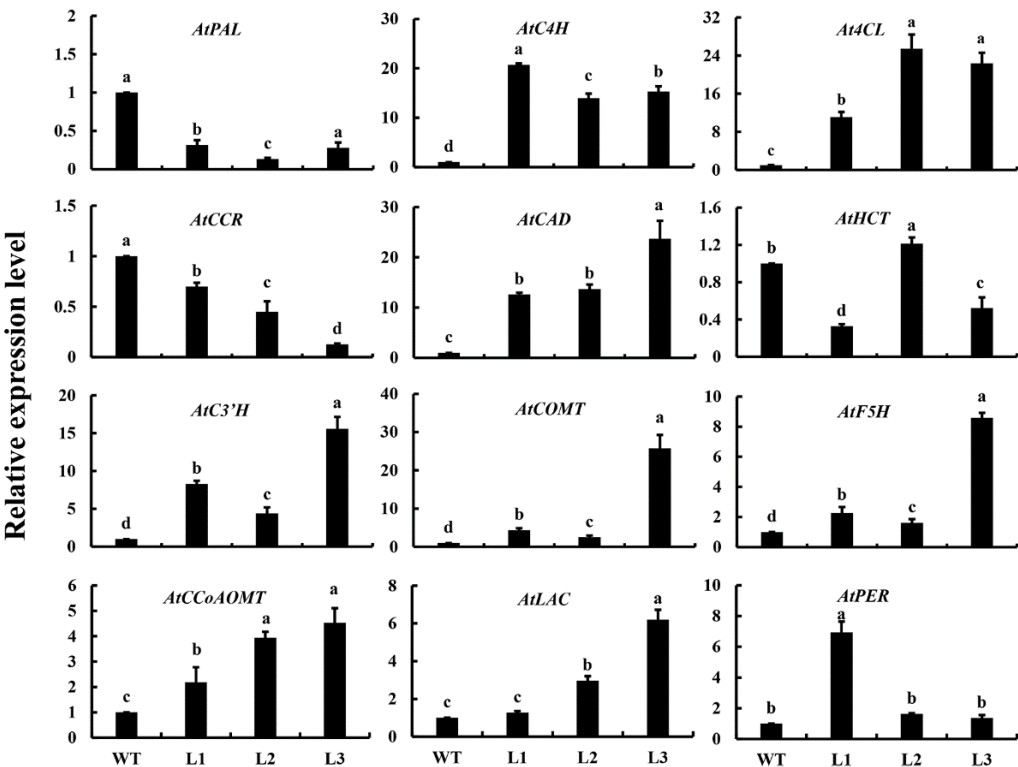

**Figure 6.** Expression profiles of lignin-related structural genes in *A. thaliana* plants. Data are expressed as the means ± standard deviation (SD) of three replicates. Different lowercase letters indicate significant differences at 0.05 levels.

## 4. Discussion

Celery is a popular vegetable crop in China and around the world, and its leaf blades and petioles contain rich nutrients that humans need. Celery is rich in dietary fiber and contains more lignin than other vegetables. The biosynthesis of lignin plays an important role in the growth and development of plants that not only "affects" the taste and flavor of vegetable crops, but also supports plant structure and promotes disease resistance in plants, such as carrots [35], *Toona sinensis* [36], tea [13], and maize [37]. Appropriate amounts of lignin support green and straight celery with a delicious taste, which is beneficial for improving the quality and yield of celery. In our study, we found that as the lignin content of celery petioles increased, the celery tissues became lignified and inedible. Therefore, studying the regulation of lignin synthesis during the growth and development of celery is crucial for the quality of celery.

Regarding the lignin metabolism pathway, the first step is the formation of aromatic amino acids through the shikimic acid pathway, and the next step is hydroxycinnamic acids and hydroxycinnamoyl CoA compounds, which are synthesized from aromatic amino acids through deamination, hydroxylation, and methylation. The genes involved in lignin biosynthesis have been validated in multiple species, such as *Eucalyptus* [38], *Acacia auriculiformis*, *Acacia mangium* [39], and Chinese fir [40]. With many years of research, the key enzymes involved in regulating lignin biosynthesis pathways currently include phenylalanine ammonia lyase (PAL), cinnamate 4-hydroxylase (C4H), 4-coumarate-CoA ligase (4CL), cinnamoyl-CoA reductase (CCR), cinnamyl alcohol dehydrogenase (CAD), hydroxycinnamoyl-CoA shikimate/quinate hydroxycinnamoyl transferase (HCT), p-coumaroyl 3′-hydroxylase (C3′H), caffeoyl-CoA O-methyltransferase (CCoAOMT), ferulate 5-hydroxylase (F5H), caffeic acid O-methyltransferase (COMT), laccase (LAC), and peroxidase (PER) [41–43].

4CL is a key enzyme involved in lignin biosynthesis, which catalyzes the conversion of 4-cinnamic acid to 4-cinnamoyl CoA, and participates in the biosynthesis of other secondary

metabolites [44]. *4CL* has been confirmed to exist in various plants, and its expression is tissue specific. Five 4CLs were found in rice [45], of which *Os4CL2* was specifically expressed in anthers and strongly activated by ultraviolet radiation, indicating that it may be involved in the formation of flavonoids. By contrast, the remaining *4CLs* (*Os4CL1*, *Os4CL3, Os4CL4, Os4CL5*) were involved in lignin formation [46]. Twelve *Pg4CLs* were identified in the pomegranate genome [47], and *Pg4CL5* successfully clustered with *At4CL1*, *At4CL2*, and *At4CL4*, indicating that *Pg4CL5* belonged to class I, which was involved in lignin synthesis; *Pg4CL2* clustered with *At4CL3* and belonged to class II, which was related to flavonoid synthesis.

Antisense inhibits the expression of *Os4CL3* in rice; the plants become dwarfed, and the content of coumaric acid and ferulic acid increases, whereas the lignin content decreases, resulting in plant anther dysplasia. Moreover, the fertility and yield of transgenic rice decreased [24,48]. In Fraxinus mandsshurica [49], the Fm4CL2-overexpressing (OE-Fm4CL2) tobacco showed increased lignin content and decreased hemicellulose content. In addition, silencing *Gh4CL7* in upland cotton resulted in a decrease of approximately 20% in lignin content compared with the control group, whereas overexpressing *Gh4CL7* in transgenic *A. thaliana* OE lines increased the lignin content by approximately 10% [50].

Previous studies have shown that changes in lignin content are closely associated with the expression of lignin biosynthesis genes [51,52]. In the lignin biosynthesis pathway, *PAL*, *C4H*, *4CL*, *CCR*, *CAD*, *HCT*, *C3′H*, *COMT*, *F5H*, *CCoAOMT*, *LAC*, and *PER* are structural genes that directly affect lignin formation through enzyme catalysis. *PAL* and *C4H* are not specifically involved in lignin synthesis, but they are involved in the synthesis of non-lignin phenolic compounds, such as flavonoids and salicylic acid. *PAL* and *C4H* gene expression was inhibited, their activity was controlled, and the precursor required for lignin monomer synthesis upstream of the lignin synthesis pathway was reduced, thereby affecting lignin synthesis [53]. The overexpression or inhibition of the *CCR* gene not only affects transgenic plant lignin content, but also affects its normal growth and development [54–57]. The activity of the *CAD* gene was inhibited in certain plants, and the lignin content remained unchanged [58,59]. The overexpression of the *F5H* gene in tobacco, poplar, and *Arabidopsis* significantly increased S-lignin biosynthesis, whereas G-lignin biosynthesis was significantly inhibited [60–62]. Reducing the *COMT* gene caused the lignin content to decrease in transgenic alfalfa [63] and poplar [64]. In transgenic alfalfa, the activity of the *CCoAOMT* gene was associated with G-lignin content [65]. The inhibition of the *C3H* gene in transgenic alfalfa did not significantly change the lignin content, but the H-lignin content increased by approximately 65% relative to the S-lignin and G-lignin contents [66,67]. Silencing *AtHCT* gene expression inhibited lignin synthesis, and the height of transgenic *A. thaliana* plants was significantly reduced [68]. The overexpression of *4CL* genes can affect the expression of downstream genes in lignin, such as *CCR*, *CAD*, *HCT*, *C3′H*, *COMT*, *F5H*, *CCoAOMT*, *LAC*, and *PER*. In this study, the overexpression of *Ag4CL3* led to a significant upregulation of lignin biosynthesis gene expression in transgenic *A. thaliana* plants, except for *AtPAL*, *AtCCR*, and *AtLAC*.

In our investigation, we cloned the *Ag4CL3* gene sequence approximately 1668 bp fragment successfully from *Apium graveolens*. Moreover, during the growth stages of celery, the expression level of *Ag4CL3* gradually increased in the petioles and leaf blades, which is consistent with the increasing lignin accumulation at the three stages. Therefore, this gene may be involved in the biosynthetic pathway of lignin. This finding is consistent with the expression results in *Populus tremuloide* [26] and *Nicotiana tabacum* [69]. To further discuss the function of the *Ag4CL3* gene, we overexpressed the *Ag4CL3* gene and measured the lignin content of three *A. thaliana* OE lines and WT. Compared with the WT, the overexpression of *Ag4CL3* upregulated the lignin content in three transgenic *A. thaliana* lines. Meanwhile, the overexpression of *Ag4CL3* led to the lignin biosynthesis gene expression in the transgenic *A. thaliana* plants being significantly upregulated, except for *AtPAL*, *AtCCR*, and *AtLAC*. The overexpression of *Ag4CL3* can possibly improve the activity of 4-Coumarate: coenzyme A ligase, which is involved in the lignin biosynthesis pathway in

*A. thaliana*. With the published celery genome and transcriptomic data, we found five *4CL* genes in celery. In this study, only the *Ag4CL3* gene was studied, and the function of the remaining four genes needs further investigation.

## 5. Conclusions

4-Coumarate: coenzyme A ligase is an important enzyme in the lignin biosynthesis pathway. In this study, we cloned the *Ag4CL3* gene sequence successfully from *Apium graveolens*. The *Ag4CL3* gene contains an ORF with a length of 1688 bp, 555 amino acids. The overexpression of *Ag4CL3* led to the lignin biosynthesis gene expression in the transgenic *A. thaliana* plants being almost significantly upregulated, except for *AtPAL, AtCCR,* and *AtLAC*. This study speculates that *Ag4CL3* genes are related to lignin synthesis in *A. graveolens*.

**Author Contributions:** Conceptualization, X.-L.Z., S.-H.Z. and G.-F.T.; Methodology, X.-L.Z., S.-H.Z. and Q.Z.; Software, X.-L.Z. and Q.Z.; Validation, X.-L.Z. and G.-F.T.; Formal analysis, S.-H.Z., Q.L., K.W. and Z.-F.C.; Data curation and writing—original draft preparation, X.-L.Z. and G.-F.T.; Writing—review and editing, G.-F.T.; Supervision, G.-F.T.; Project administration, X.-L.Z. and G.-F.T.; Visualization, X.-L.Z. All authors have read and agreed to the published version of the manuscript.

**Funding:** This study was financially supported by Guizhou Province Basic Research Program ([2019]1307); Guizhou Academy of Agricultural Sciences Support Project [Qian Nongkeyuan Science and Technology Innovation No. (2022) 07]; Project of Guizhou Provincial Department of Science and Technology (No. Qiankehe Fuqi [2022] 005); Vegetable System Project of Guizhou (GZCYTX2023-0101); and Construction of Guiyang Vegetable Germplasm Resources Research Center (Zhuke contract [2021] No. 5-1).

**Data Availability Statement:** Not applicable.

**Conflicts of Interest:** The authors declare no conflict of interest.

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
