# Peer review of "Ag4CL3 Related to Lignin Synthesis in Apium graveolens L."

_agronomy, doi:10.3390/agronomy13082025_

Round 1
Reviewer 1 Report
The manuscript entitled” Ag4CL3 Related to Lignin Synthesis in Apium graveolens L.” authored by Zhong et al. is interesting. I have read the manuscript and suggest some minor revisions before accepting for publication.
1. The experimental plan is a reasonably design in this paper, and language clearly and smoothly. The authors should carefully check the language.
2. Phylogenetic analysis showed that the Ag4CL3 protein had the closest evolutionary relationship with carrot. Spell out the Latin name for carrot here.
3. The authors need to improve the quality of the figures, such as Figure 3. Also more detailed legend should be added to fig 3, the number should be either on above the branch or below the branch, making consistency, what the number stands for should be added.
4. In Figure 5B, the authors need to add the abscissa label in this figure.
5. In section 3.5: Ag4CL3 is mainly involved in lignin biosynthesis and determines the expression. From Fig 7, I cannot find evidence to support this statement. More information needs to explain this.
6. of lignin-related genes.
7. In identification of transgenic plants A. thaliana, the authors should explain clearly which pair of the primers were used for PCR identification.
8. Check the manuscript for italics, some words should be italics such as transgenic etc.
9. Which author belongs to the Institute 4?
The authors should carefully check the language
Author Response
22 Jul, 2023
Manuscript ID: agronomy- 2474060
Title: Ag4CL3 Related to Lignin Synthesis in Apium graveolens L.
Journal name: Agronomy
Manuscript ID: agronomy- 2474060
Journal: Agronomy
Author: Xiu-Lai Zhong, Shun-Hua Zhu, Qian Zhao, Qing Luo, Kun Wang, Zhi-Feng Chen *,
Guo-Fei Tan *
Dear Editors and Reviewers,
Thank you and the reviewers very much for revising our manuscript ‘Ag4CL3 Related to Lignin Synthesis in Apium graveolens L.’ (MANUSCRIPT ID: agronomy- 2474060). Your effort and time spent on our manuscript are greatly appreciated by all of us. We are delighted to all suggestion and review comments, which you and the reviewers made. Your revisions/suggestions have definitely improved the quality of our manuscript.
Please find the revised manuscript in Agronomy’s manuscript center. The changes were made directly in the text with Trace or Blue marked. The responses to the reviewers are highlighted below.
Thank you again for your kind help and excellent suggestions for our manuscript. We hope these revisions will be satisfactory. We are looking forward to hearing from you soon.
Yours sincerely
Dr. Guo-Fei Tan, tagfei@foxmail.com
Institute of Horticulture, Guizhou Academy of Agricultural Sciences
Guiyang, China
Comments from the editors and reviewers:
Reviewer #1:
- Phylogenetic analysis showed that the Ag4CL3 protein had the closest evolutionary relationship with carrot. Spell out the Latin name for carrot here.
Response:
-- We thank the reviewer for this observation and suggestion.
-- We agree. We appreciate your comments, we have modified the sentence as follows:
- The authors need to improve the quality of the figures, such as Figure 3. Also more detailed legend should be added to fig 3, the number should be either on above the branch or below the branch, making consistency, what the number stands for should be added.
Response:
-- We thank the reviewer for this observation and suggestion.
-- We agree. We appreciate your comments, we have modified the sentence and figure as follows:
- In Figure 5B, the authors need to add the abscissa label in this figure.
Response:
-- We thank the reviewer for this observation and suggestion.
-- We agree. We appreciate your comments, we have modified the figure as follows:
- In section 3.5: Ag4CL3 is mainly involved in lignin biosynthesis and determines the expression. From Fig 7, I cannot find evidence to support this statement. More information needs to explain this.
Response:
-- We thank the reviewer for this observation and suggestion.
-- We agree. We appreciate your comments, we have modified the sentence as follows:
- of lignin-related genes.
Response:
-- We thank the reviewer for this observation and suggestion.
-- We agree. We appreciate your comments, we have modified the sentence as follows:
- In identification of transgenic plants A. thaliana, the authors should explain clearly which pair of the primers were used for PCR identification.
Response:
-- We thank the reviewer for this observation and suggestion.
-- We agree. We appreciate your comments, we have modified the sentence as follows:
- Check the manuscript for italics, some words should be italics such as transgenic etc.
Response:
-- We thank the reviewer for this observation and suggestion.
-- We agree. We appreciate your comments, we have checked and modified the words as follows:
- Which author belongs to the Institute 4?
Response:
-- We thank the reviewer for this observation and suggestion.
-- We agree. We appreciate your comments, we have modified the sentence as follows:

Reviewer 2 Report
Zhong et al studied the role of Ag4CL3 in lignin synthesis in celery. The authors cloned the Ag4CL3 gene, examined gene expression levels and lignin contents during the development of celery, and created Ag4CL3 over-expression Arabidopsis mutants. The experimental design is valid, but the conclusions are not fully supported by evidence.
Specific comments:
- The writing needs improvement. Some sentences are understandable, but not easy to read or can cause confusion. Below are places that can be improved:
- ‘4-Coumarate: coenzyme A ligase (4CL; EC 6.2.1.12) is an important enzyme…’ should be changed to ‘4-Coumarate, coenzyme A ligase (4CL; EC 6.2.1.12), is an important enzyme…’
- ‘the 4CL proteins from celery and carrot belonged to the same branch’ change ‘branch’ to ‘clade’.
- ‘the overexpression of Ag4CL3 led to the almost significant upregulation of lignin biosynthesis gene’. What is ‘almost significant’? If it is statistically significant, then remove ‘almost’. If it is not statistically significant, avoid using the word ‘significant’. Same for section 3.5, the result section, and the conclusion section. Please avoid using ‘almost significant’.
- ‘expect for AtPAL, AtCCR and AtLAC’. I do not understand this part of the sentence. Were up-regulation of these 3 gene expected? Or do the authors mean ‘except’? Some word confusion exists in section 3.5, the result section, and the conclusion section.
- ‘In celery, the function of the Ag4CL gene lignin biosynthesis is unclear.’ Need to add ‘in/ during’ between ‘gene’ and ‘lignin’.
- ‘Transgenic A. thaliana plants overexpressing Ag4CL3 were obtained to examine the lignin level and then compared.’ First, ‘Transgenic’ should not be italicized. Second, then compared to what? Please finish the sentence.
- ‘At 70 d after sowing, the Ag4CL3 gene could be expressed in the roots, leaves, and petioles’. What does ‘could be’ imply? Is it expressed or not?
- ‘was highest in petioles and second roots and lowest in leaves (Figure 4A)’. Add ‘in’ between ‘second’ and ‘roots’.
- Figure 5 caption: ‘Dates are expressed as the …’ Does the authors mean ‘data’?
- ‘other lignin biosynthesis genes was detected in transgenic A. thaliana (Figure 7).’. Change ‘detected’ to ‘examined/ investigated/ measured’ etc.
- ‘Despite many years of research, the key enzymes involved in regulating lignin biosynthesis pathways currently include…’ I don’t understand the transition here, ‘despite of what?’
- Please add more details in the section 2.4 Bioinformatics Analysis
- How were the homologous sequences from other species identified from NCBI? Is it a BLAST search or a keyword search or a domain search? How can the authors confirm that there aren’t other copies of 4CL3 in these species? If the selection is based on literature, please add references.
- Which method was used to construct the phylogenetic tree? From the results section, I did figure out that Neighbor joining was used to infer the phylogeny, but the authors need to have this information here as well. In addition, what parameters were used? Is there a sequence alignment step before constructing the phylogenetic tree? If so, what method is used?
- What parameters were used to run the other softwares? If default parameters were used, please note.
- Were there technical replicates in the qPCR experiments?
- Figure 2 should be a supplementary figure. Information from this figure is not directly linked to the story. For example, if the authors studied the role of a specific phosphorylation site, then figure 2E is essential; if the author compare function of genes with or without the signal peptide, then figure 2D is necessary. Since none of the information presented in Figure 2 is related to downstream analysis, this figure should be moved to supplementary materials.
- Please use a Maximum likelihood method or bayesian method to build the phylogenetic tree. Neighbor joining is a clustering method, rather than a phylogenetic inference method. Both methods are implemented in MEGA and should be easy to execute.
- Figure 4A and 4B should be switched. The figure is discordant with the text and figure caption.
- ‘The Ag4CL3 protein is highly conserved among different plant species.’ I see no evidence supporting this claim. Authors need to add alignment and domain analysis to support this claim.
- ‘These findings suggest that Ag4CL3 acted as a regulator of lignin biosynthesis and growth and development in celery.’ Is a growth and development regulator of the whole plant, including roots and flowers? Or just a regulator of leave and petiole development? Also I don’t think the authors provided enough evidence to support this claim for the following reasons: 1) The evidence is over-expression in Arabidopsis, thus it can not be directly translated to celery; 2) there is no growth assay in the over-expression lines.
Please see specific comments 1. I've listed a few places that needs improvement, but please read through the whole manuscript and make changes as needed.
Author Response
22 Jul, 2023
Manuscript ID: agronomy- 2474060
Title: Ag4CL3 Related to Lignin Synthesis in Apium graveolens L.
Journal name: Agronomy
Manuscript ID: agronomy- 2474060
Journal: Agronomy
Author: Xiu-Lai Zhong, Shun-Hua Zhu, Qian Zhao, Qing Luo, Kun Wang, Zhi-Feng Chen *,
Guo-Fei Tan *
Dear Editors and Reviewers,
Thank you and the reviewers very much for revising our manuscript ‘Ag4CL3 Related to Lignin Synthesis in Apium graveolens L.’ (MANUSCRIPT ID: agronomy- 2474060). Your effort and time spent on our manuscript are greatly appreciated by all of us. We are delighted to all suggestion and review comments, which you and the reviewers made. Your revisions/suggestions have definitely improved the quality of our manuscript.
Please find the revised manuscript in Agronomy’s manuscript center. The changes were made directly in the text with Trace or Blue marked. The responses to the reviewers are highlighted below.
Thank you again for your kind help and excellent suggestions for our manuscript. We hope these revisions will be satisfactory. We are looking forward to hearing from you soon.
Yours sincerely
Dr. Guo-Fei Tan, tagfei@foxmail.com
Institute of Horticulture, Guizhou Academy of Agricultural Sciences
Guiyang, China
Comments from the editors and reviewers:
Reviewer #2:
- ‘4-Coumarate: coenzyme A ligase (4CL; EC 6.2.1.12) is an important enzyme…’ should be changed to ‘4-Coumarate, coenzyme A ligase (4CL; EC 6.2.1.12), is an important enzyme…’
Response:
-- We thank the reviewer for this observation and suggestion.
-- We agree. We appreciate your comments, we have modified the sentence as follows:
- ‘the 4CL proteins from celery and carrot belonged to the same branch’ change ‘branch’ to ‘clade’.
Response:
-- We thank the reviewer for this observation and suggestion.
-- We agree. We appreciate your comments, we have modified the sentence as follows:
- ‘the overexpression of Ag4CL3 led to the almost significant upregulation of lignin biosynthesis gene’. What is ‘almost significant’? If it is statistically significant, then remove ‘almost’. If it is not statistically significant, avoid using the word ‘significant’. Same for section 3.5, the result section, and the conclusion section. Please avoid using ‘almost significant’.
Response:
-- We thank the reviewer for this observation and suggestion.
-- We agree. We appreciate your comments, we have modified the sentence as follows:
Abstract:
Section 3.5:
Result section:
Conclusion section:
- ‘expect for AtPAL, AtCCR and AtLAC’. I do not understand this part of the sentence. Were up-regulation of these 3 gene expected? Or do the authors mean ‘except’? Some word confusion exists in section 3.5, the result section, and the conclusion section.
Response:
-- We thank the reviewer for this observation and suggestion.
-- We agree. We appreciate your comments, we have modified the sentence as follows:
Abstract:
Section 3.5:
Result section:
Conclusion section:
- ‘In celery, the function of the Ag4CL gene lignin biosynthesis is unclear.’ Need to add ‘in/ during’ between ‘gene’ and ‘lignin’.
Response:
-- We thank the reviewer for this observation and suggestion.
-- We agree. We appreciate your comments, we have modified the sentence as follows:
- ‘Transgenic A. thaliana plants overexpressing Ag4CL3 were obtained to examine the lignin level and then compared.’ First, ‘Transgenic’ should not be italicized. Second, then compared to what? Please finish the sentence.
Response:
-- We thank the reviewer for this observation and suggestion.
-- We agree. We appreciate your comments, we have modified the sentence as follows:
- ‘At 70 d after sowing, the Ag4CL3 gene could be expressed in the roots, leaves, and petioles’. What does ‘could be’ imply? Is it expressed or not?
Response:
-- We thank the reviewer for this observation and suggestion.
-- We agree. We appreciate your comments, we have modified the sentence as follows:
- ‘was highest in petioles and second roots and lowest in leaves (Figure 4A)’. Add ‘in’ between ‘second’ and ‘roots’.
Response:
-- We thank the reviewer for this observation and suggestion.
-- We agree. We appreciate your comments, we have modified the sentence as follows:
- Figure 5 caption: ‘Dates are expressed as the …’ Does the authors mean ‘data’?
Response:
-- We thank the reviewer for this observation and suggestion.
-- We agree. We appreciate your comments, we have modified the sentence as follows:
- ‘other lignin biosynthesis genes was detected in transgenic A. thaliana (Figure 7).’. Change ‘detected’ to ‘examined/ investigated/ measured’ etc.
Response:
-- We thank the reviewer for this observation and suggestion.
-- We agree. We appreciate your comments, we have modified the sentence as follows:
- ‘Despite many years of research, the key enzymes involved in regulating lignin biosynthesis pathways currently include…’ I don’t understand the transition here, ‘despite of what?’
Response:
-- We thank the reviewer for this observation and suggestion.
-- We agree. We appreciate your comments, we have modified the sentence as follows:
- Please add more details in the section 2.4 Bioinformatics Analysis.
1) How were the homologous sequences from other species identified from NCBI? Is it a BLAST search or a keyword search or a domain search? How can the authors confirm that there aren’t other copies of 4CL3 in these species? If the selection is based on literature, please add references.
2) Which method was used to construct the phylogenetic tree? From the results section, I did figure out that Neighbor joining was used to infer the phylogeny, but the authors need to have this information here as well. In addition, what parameters were used? Is there a sequence alignment step before constructing the phylogenetic tree? If so, what method is used?
3) What parameters were used to run the other softwares? If default parameters were used, please note.
Response:
-- We thank the reviewer for this observation and suggestion.
-- We agree. We appreciate your comments, we have modified the sentence as follows:
- Were there technical replicates in the qPCR experiments?
Response:
-- We thank the reviewer for this observation and suggestion.
-- There are three technical replicates in the qPCR experiments.
- Figure 2 should be a supplementary figure. Information from this figure is not directly linked to the story. For example, if the authors studied the role of a specific phosphorylation site, then figure 2E is essential; if the author compare function of genes with or without the signal peptide, then figure 2D is necessary. Since none of the information presented in Figure 2 is related to downstream analysis, this figure should be moved to supplementary materials.
Response:
-- We thank the reviewer for this observation and suggestion.
-- We agree. We appreciate your comments, we have modified the sentence and figure as follows:
- Please use a Maximum likelihood method or bayesian method to build the phylogenetic tree. Neighbor joining is a clustering method, rather than a phylogenetic inference method. Both methods are implemented in MEGA and should be easy to execute.
Response:
-- We thank the reviewer for this observation and suggestion.
-- We agree. We appreciate your comments, we have modified the sentence as follows:
- Figure 4A and 4B should be switched. The figure is discordant with the text and figure caption.
Response:
-- We thank the reviewer for this observation and suggestion.
-- We agree. We appreciate your comments, we have modified the figure as follows:
- ‘The Ag4CL3 protein is highly conserved among different plant species.’ I see no evidence supporting this claim. Authors need to add alignment and domain analysis to support this claim.
Response:
-- We thank the reviewer for this observation and suggestion.
-- We agree. We appreciate your comments, we have modified the sentence as follows:
- ‘These findings suggest that Ag4CL3 acted as a regulator of lignin biosynthesis and growth and development in celery.’ Is a growth and development regulator of the whole plant, including roots and flowers? Or just a regulator of leave and petiole development? Also I don’t think the authors provided enough evidence to support this claim for the following reasons: 1) The evidence is over-expression in Arabidopsis, thus it can not be directly translated to celery; 2) there is no growth assay in the over-expression lines.
Response:
-- We thank the reviewer for this observation and suggestion.
-- We agree. We appreciate your comments, we have modified the sentence as follows:
Abstract:
Conclusions:

Reviewer 3 Report
This paper by Zhong et al., is a standard study of the expression of a cloned gene. In this case the gene is from celery and is part of the lignin biosynthetic pathway. I have only a few minor comments on the content and the English.
Figure 2 could be condensed. Parts B, D, and E could be deleted without affecting the message so that the other parts of figure 2 could be enlarged.
There is no figure 6.
They make the comment on page 6 that no signal sequence is found. Where is the enzyme likely localized?
For the English comments, since there are no line numbers, it is difficult to describe the place where corrections need to be made.
First, when describing upregulated genes in transgenic Arabidopsis, every time the word used is “expect”. However, the word should be “except”. Pages 1, 11 (twice in last 2 paragraphs), and Conclusions.
Be consistent with capital letters in your headings. For example: 3.1 Changes of Lignin Content has all words capitalized that should be. However, in 3.2, Sequence should also be capitalized. I would also add the word “the” in front of the gene name: “the Ag4CL3 Sequence”.
Page 2, second paragraph: sinapoly should be sinapol.
Page 2, 4th paragraph: first line: the Ag4CL gene “in” lignin biosynthesis . . .
In several figure legends, the authors say “Different lowercase letters indicates”. The word should be “indicate”. (No S).
Top of page 8: “could be” should be “was”
Discussion:
First paragraph: the last sentence is confusing. I would suggest the following: The biosynthesis of lignin plays an important role in the growth and development of plants that not only “affects” the taste and flavor of vegetable crops, but also support plant structure and promotes disease resistance in plants, etc.
Third paragraph: The authors say “despite many years of research”. Why “despite”, how about “Because of”?
Page 11, first paragraph: flavonoids. By contrast (end a sentence and begin a new one.)
Also, “12 Pg4CLs were identified, and Pg4CL4 was found to be highly expressed in the , , ,” (sentence was awkward).
Second paragraph: The sentence on Manchurian ash doesn’t make sense—Did the citation show the Manchurian ash gene in tobacco? If so, please reword.
Page 11, 8 lines from the bottom: This finding “is” consistent.
Conclusions:
First line: “lignin” is missing the “l”
Good
Author Response
22 Jul, 2023
Manuscript ID: agronomy- 2474060
Title: Ag4CL3 Related to Lignin Synthesis in Apium graveolens L.
Journal name: Agronomy
Manuscript ID: agronomy- 2474060
Journal: Agronomy
Author: Xiu-Lai Zhong, Shun-Hua Zhu, Qian Zhao, Qing Luo, Kun Wang, Zhi-Feng Chen *,
Guo-Fei Tan *
Dear Editors and Reviewers,
Thank you and the reviewers very much for revising our manuscript ‘Ag4CL3 Related to Lignin Synthesis in Apium graveolens L.’ (MANUSCRIPT ID: agronomy- 2474060). Your effort and time spent on our manuscript are greatly appreciated by all of us. We are delighted to all suggestion and review comments, which you and the reviewers made. Your revisions/suggestions have definitely improved the quality of our manuscript.
Please find the revised manuscript in Agronomy’s manuscript center. The changes were made directly in the text with Trace or Blue marked. The responses to the reviewers are highlighted below.
Thank you again for your kind help and excellent suggestions for our manuscript. We hope these revisions will be satisfactory. We are looking forward to hearing from you soon.
Yours sincerely
Dr. Guo-Fei Tan, tagfei@foxmail.com
Institute of Horticulture, Guizhou Academy of Agricultural Sciences
Guiyang, China
Reviewer #3:
- Figure 2 could be condensed. Parts B, D, and E could be deleted without affecting the message so that the other parts of figure 2 could be enlarged.
Response:
-- We thank the reviewer for this observation and suggestion.
-- We agree. We appreciate your comments, we have modified the sentence and figure as follows:
- There is no figure 6.
Response:
-- We thank the reviewer for this observation and suggestion.
-- We agree. We appreciate your comments, we have modified the figure as follows:
- They make the comment on page 6 that no signal sequence is found. Where is the enzyme likely localized?
Response:
-- We thank the reviewer for this observation and suggestion.
-- We agree. We appreciate your comments, we have modified the sentence as follows:
- For the English comments, since there are no line numbers, it is difficult to describe the place where corrections need to be made.
Response:
-- We thank the reviewer for this observation and suggestion.
-- We agree. We appreciate your comments, We added the line numbers to the right of the article:
- First, when describing upregulated genes in transgenic Arabidopsis, every time the word used is “expect”. However, the word should be “except”. Pages 1, 11 (twice in last 2 paragraphs), and Conclusions.
Response:
-- We thank the reviewer for this observation and suggestion.
-- We agree. We appreciate your comments, we have modified the sentence as follows:
Abstract:
Section 3.5:
Result section:
Conclusion section:
- Be consistent with capital letters in your headings. For example: 3.1 Changes of Lignin Content has all words capitalized that should be. However, in 3.2, Sequence should also be capitalized. I would also add the word “the” in front of the gene name: “the Ag4CL3 Sequence”.
Response:
-- We thank the reviewer for this observation and suggestion.
-- We agree. We appreciate your comments, we have modified the headings as follows:
- Page 2, second paragraph: sinapoly should be sinapol.
Response:
-- We thank the reviewer for this observation and suggestion.
-- We agree. We appreciate your comments, we have modified the sentence as follows:
- Page 2, 4th paragraph: first line: the Ag4CL gene “in” lignin biosynthesis . . .
Response:
-- We thank the reviewer for this observation and suggestion.
-- We agree. We appreciate your comments, we have modified the sentence as follows:
- In several figure legends, the authors say “Different lowercase letters indicates”. The word should be “indicate”. (No S)
Response:
-- We thank the reviewer for this observation and suggestion.
-- We agree. We appreciate your comments, we have modified the figure and table legends as follows:
- Top of page 8: “could be” should be “was”
Response:
-- We thank the reviewer for this observation and suggestion.
-- We agree. We appreciate your comments, we have modified the sentence as follows:
- Discussion:
First paragraph: the last sentence is confusing. I would suggest the following: The biosynthesis of lignin plays an important role in the growth and development of plants that not only “affects” the taste and flavor of vegetable crops, but also support plant structure and promotes disease resistance in plants, etc.
Third paragraph: The authors say “despite many years of research”. Why “despite”, how about “Because of”?
Response:
-- We thank the reviewer for this observation and suggestion.
-- We agree. We appreciate your comments, we have modified the sentence as follows:
First paragraph:
Third paragraph:
- Page 11,
First paragraph: flavonoids. By contrast (end a sentence and begin a new one.). Also, “12 Pg4CLs were identified, and Pg4CL4 was found to be highly expressed in the , , ,” (sentence was awkward).
Second paragraph: The sentence on Manchurian ash doesn’t make sense—Did the citation show the Manchurian ash gene in tobacco? If so, please reword.
Response:
-- We thank the reviewer for this observation and suggestion.
-- We agree. We appreciate your comments, we have modified the sentence as follows:
First paragraph:
Second paragraph:
- Page 11, 8 lines from the bottom: This finding “is” consistent.
Response:
-- We thank the reviewer for this observation and suggestion.
-- We agree. We appreciate your comments, we have modified the sentence as follows:
40.Conclusions: First line: “lignin” is missing the “l”
Response:
-- We thank the reviewer for this observation and suggestion.
-- We agree. We appreciate your comments, we have modified the sentence as follows:
